# Analysis of the Consumption of Sports Supplements in Open Water Swimmers According to the Competitive Level

**DOI:** 10.3390/nu14245211

**Published:** 2022-12-07

**Authors:** Rubén Jiménez-Alfageme, Raúl Domínguez, Antonio Jesús Sanchez-Oliver, Paola Tapia-Castillo, José Miguel Martínez-Sanz, Isabel Sospedra

**Affiliations:** 1Faculty of Health Sciences, University of Alicante, 03690 Alicante, Spain; 2Food and Nutrition Research Group (ALINUT), University of Alicante, 03690 Alicante, Spain; 3Physiotherapy Department, Faculty of Health Sciences, European University of Gasteiz—EUNEIZ, La Biosfera Ibilbidea, 6, 01013 Vitoria-Gasteiz, Spain; 4Studies Research Group in Neuromuscular Responses (GEPREN), University of Lavras, Lavras 37203-202, Brazil; 5Departamento de Motricidad Humana y Rendimiento Deportivo, Facultad de Ciencias de la Educación, Universidad de Sevilla, 41004 Sevilla, Spain; 6Ministerio del Deporte, Ibarra 170505, Ecuador; 7Nursing Department, Faculty of Health Sciences, University of Alicante, 03690 Alicante, Spain

**Keywords:** sports supplements, open water swimming, sport nutrition, ergogenic aids, performance

## Abstract

Background: Sports supplements (SS) are widely consumed by many types of athletes to improve their performance. These SS are classified according to their level of scientific evidence, by the ABCD system from the Australian Institute of Sport (AIS). In open water swimming, their use may be necessary due to the physiological challenges posed by this sport discipline. However, there is currently little literature on the use of SS in open water swimmers. The aim of this work is to analyze the pattern of consumption of SS by open water swimmers, by studying the differences according to the competitive level (regional vs. national). Methods: Descriptive and cross-sectional study on the consumption and use of SS by federated open water swimmers in Spain in the 2019/2020 and 2020/2021 seasons. The data were collected through a validated questionnaire. Results: 79.5% of the participants consumed SS, with significant differences according to their level, being higher in athletes at the national level. The most-consumed SS by the swimmers studied were sports drinks, energy bars, caffeine, vitamin C, and vitamin D. Conclusions: It was observed that the consumption of SS in open water swimmers was high, and of the five most-consumed SS, four of them belonged to the category of greatest scientific evidence.

## 1. Introduction

Presently, between 40–70% of athletes of all levels consume sports supplements (SS) to improve their sports performance [1]. SS can be defined as a food, component, nutrient, or non-food component that is ingested within a normal diet in order to obtain a certain effect on health or performance [2]. Regarding the differences according to sex and competitive level, the literature shows that men tend to consume more SS than women, as well as elite athletes more than amateurs [2,3].

Although there is a wide variety of supplements on the market, it is necessary to differentiate them according to their scientific evidence of improved sports performance. Thus, the Australian Institute of Sport (AIS) classifies them into four groups (ABCD) from a greater to lesser degree of scientific evidence [4].

Within endurance sports, we find open water swimming (OWS), which has experienced a surge in both competitive tests and participants in the last few decades [5]. Since the Olympic Games in Beijing 2008, the OWS modality was incorporated into swimming competitions. Olympic distance in OWS is called “Marathon” of 10 km, and international competitions include distances of 5, 10, and 25 km [6]. This discipline creates unique physiological challenges for thermoregulation and muscle fuel reserves, due to the water temperature and the duration of some competitions, which can reach 5 or 6 h [5]. These high physiological demands, as well as the possibility that any small gain could result in an improvement in performance and competition, encourages athletes to consider the use of different tools and/or strategies, among which we find the use of sports supplements [7]. In addition, in OWS, the supply of nutrients during competition takes place from floating stations, where swimmers take food from the end of a pole, at the same time that they slow down, posing another challenge of the sport discipline [8]. Reduced swimming speed for food replenishment and SS can cause slower swimmers in this process to be overtaken by their competitors.

In endurance sports, the current literature suggests that the most-consumed SS are caffeine, which benefits performance [9]; and energy drinks, which have demonstrated benefits in sports performance by compensating for the loss of fluids and nutrients during training or competition [10]. In addition, many athletes make use of sports gels and carbohydrate drinks during the competition, as their consumption increases endurance capacity [11].

However, although a study was found [8] that specifically described optimal nutrition for OWS swimmers, including the consumption of SS such as caffeine and possibly beta-alanine, as interesting ergogenic aids for these athletes; to date, no work analyzing the consumption of SS by open water swimmers has been found. There are some studies on SS consumption in a variety of sports, including endurance sports [3,12,13], but this is pioneering research in this population group.

Therefore, the aim of this original investigation is to analyze, using an online questionnaire, the number and the type of sport supplements consumed by open water swimmers in relation to the degree of scientific evidence and competitive level. This information will allow us to establish the sports supplements’ consumption pattern for these athletes.

## 2. Materials and Methods

### 2.1. Type of Study

A descriptive and cross-sectional study was used to investigate the consumption and regular use of SS of open water swimmers. The sample size calculation was performed with Rstudio software (version 3.15.0, Rstudio Inc., Boston, MA, USA). The significance level was set a priori at α = 0.05. The standard deviation (SD) was set according to the SS total data from previous studies in swimmers (SD = 4.00) [14]. With an estimated error (d) of 0.68, the sample size needed was 132 subjects. The study population was selected by non-probabilistic, non-injury, convenience sampling among sports federations and clubs with an open water category throughout Spain.

### 2.2. Participants and Sample Size

The study population was selected by non-probabilistic, non-injury, convenience sampling among sports federations and clubs with an open water category throughout Spain. In total, 132 federated open water swimmers participated in the 2019/2020 and 2020/2021 seasons (33.2% of the total federated swimmers in Spain). The group was composed by 103 men and 29 women, and all women compete at the regional level. All of them were adults, with a mean age of 43.11 ± 12.83 (M ± SD) years and none of the swimmers suffered any injuries or illnesses in the 6 months prior to the survey. Their competitive level differed between regional (they compete at the provincial and regional levels) and national (they compete throughout Spain). Table 1 describes the age, basic anthropometric characteristics, and years of sports experience of the study sample.

### 2.3. Instruments

The material resources used during the study were based on a questionnaire that was previously used in similar studies [7,15,16,17]. The selected supplement consumption questionnaire was validated based on content, applicability, structure, and presentation [18]. The questionnaire contained a total of 33 questions divided in three main sections: the first collected the anthropometric (e.g., age, weight, height, …), personal (e.g., sex), and social data (e.g., autonomous community of residence) of the respondent (6 questions); the second covered the practice of sports and its context (9 questions, e.g., years of practice, number of competitions, …); and the last and most extensive was related to the consumption of SS (16 questions); it can be found in: Estudio del consumo de suplementos nutricionales en nadadores de aguas abiertas (Appendix A). This part included, among other questions, what supplements they consume, why they consume them, who advises them, where they buy them, when they take them, or their perception of results after consumption.

### 2.4. Procedure

To select the sample for the study, we contacted, via email, the representatives of each regional swimming federation of Spain, as well as OWS clubs registered with these federations, to inform them about the characteristics of the study and request their collaboration. The participants’ recruitment was carried out from the same clubs and federations that informed the swimmers about the aims of the study, and after agreeing to participate, they were sent an e-mail containing the link to the SS consumption questionnaire for athletes to fill out voluntarily, electronically, and anonymously. The protocol complied with the Declaration of Helsinki for human research and was approved by the UA ethics committee with file number UA-2021-02-01.

### 2.5. Statistical Analysis

Quantitative data are expressed as mean (M) ± standard deviation (SD), and percentages as qualitative variables. A Kolmogorov–Smirnov test was used to confirm that variables were adjusted to a normal distribution, and Levene’s test was used to confirm homoscedasticity. For comparing possible differences in the number of SS consumed by national and regional swimmers, a Student’s *t*-test for independent samples was used. For comparing possible differences in the prevalence of the ingestion of SS, SS consumed with a prevalence higher than 10% and questions related to the period of consumption, timing, purpose, or motivation, the place of SS purchase or the SS advisors, a chi-square test (χ^2^) was performed. Additionally, if statistical differences were found, odds ratio (OR) and its confidence interval were applied. Statistical differences were set at *p* < 0.05. The Statistical Package for Social Sciences (version 23.0, SPSSTM Inc., Chicago, IL, USA) was used for all the statistical analyses.

## 3. Results

From the sample, 79.5% of the whole participants of the study declared consuming SS; however, a higher prevalence of consumption was found at the national as compared to the regional level (88.1% vs. 70.8%, *p* = 0.017; OR = 3.05 [1.22–7.58]). The mean number of SS ingested of the total sample was 4.67 ± 4.65, without differences between national and regional levels (5.34 ± 4.92 vs. 3.98 ± 4.28; *p* = 0.093). Table 2 shows the differences in SS consumption according to the level of evidence established by the AIS [4]. Comparing the mean intake of SS from each level, considering the AIS classification [4], a higher mean consumption of sports performance supplements (subgroup A) of swimmers at the national level (*p* = 0.027) was found, and a tendency towards statistical differences in medical supplements (subgroup A) (*p* = 0.083) and group C (*p* = 0.051), but not for sports foods (subgroup A) (*p* = 0.154), and total supplement of group A (*p* = 0.197) and group B (*p* = 0.642).

The more-consumed SS were sport drinks (62.9%), energy bars (53.0%), caffeine (39.4%), vitamin C (25.0%), and vitamin D (22.7%) (see Table 3). No statistical differences according to competition level were found for supplements in category A and B established by the AIS [4], but a higher consumption of branched-chain amino acids (BCAAs) (*p* = 0.048; OR = 2.62 [1.05–6.54]) and taurine (*p* = 0.009; OR = 6.87 [1.47–32.06]) was reported in national swimmers compared to swimmers at the regional level.

With respect to the moment of consumption, 39.4% of the swimmers declared consuming SS during the training and competition periods, 28.0% only during competition, while 11.4% consumed them exclusively during the training period. No differences were found between national and regional swimmers (*p* = 0.288). Regarding the timing of SS ingestion, 29.7% indicated pre-exercise consumption, 27.3% of the swimmers ingested them pre-, during-, and post-exercise, 25% during exercise, 7.8% post-exercise, and 10.6% indifferently. No differences were reported between competitive levels (*p* = 0.402).

When swimmers were questioned about the reason of the consumption, we based differences on competitive level (*p* = 0.009), with a higher reason for consumption based on increased sport performance (national: 52.1%, regional: 42.6%), and for health status (national: 7.0%, regional: 24.6%). In addition, differences regarding the SS purchase site were not found (*p* = 0.054), with specialized stores being the most common (27.1%), followed by pharmacies (18.1%), internet (15.7%), and mall (15.3%). Finally, no differences were found about the source of information for determining type, use, and utility of SS (*p* = 0.362). These were: trainers (18.0%) followed by dietitian–nutritionist (16.8%), teammates (16.2%), and friends (12.0%).

## 4. Discussion

The objective of this study was to analyze the consumption pattern of SS of open water swimmers, including possible differences according to their competitive level. Although we can find many studies about SS consumption in a wide variety of sports, there are only a few studies specifically in endurance athletes [19,20]; this is the first to analyze its consumption in open water swimmers. Knowing the consumption pattern of SS can contribute to improving the training and competition of these athletes, achieving a possible improvement in their athletic performance. The medical-technical staff will have the necessary information and tools to carry out a dietary-nutritional planning adjusted to the needs of the swimmers, including those SS with greater scientific evidence according to the best practice protocols.

In the present study, from the total sample, 79.5% of the swimmers surveyed consumed SS. This is a very similar result to those found in previous studies, as the consumption by ultramarathon runners (75.3%) [21], although somewhat lower than reported for mountain runners (87.5%) [20], but higher than reported by elite Spanish athletes of different modalities, including elite athletes of 21 individual sports and 7 team sports according to the following classification: endurance sport, speed/power sports, and team sports (64%) [22].

In addition, a difference in consumption was found between swimmers in different competitive levels, with a higher consumption in national-level swimmers than those who competed at the regional level, with these results similar to those previously found in studies with different Spanish athletes [17,22,23]. However, no differences were found according to sex, as in other studies [20,24] in which men consumed more SS than women.

For our sample of swimmers, the main reason for consumption of SS was the improvement in sports performance, similar to those of other Spanish athletes (45–73.2%) [20,22,23], followed by consumption to improve health status. In addition, in the present study, significant differences were found regarding motivation according to the competitive level, as proposed in previous articles [18,23].

In previous studies carried out in mountain runners, 44.4% of the athletes reported SS consumption both during training and competition, 37.5% only during the competitive period, and 8.3% during the daily training period [20], values similar to those found in our study.

As for the person who determined the consumption of SS, a determinant aspect for their correct use [13,25], in the case of swimmers, the main motivator was the trainer (18.0%), followed by the dietitian–nutritionist (16.8%), colleagues (16.2%), and friends (12%). Similar results were reported in other sports, where the majority of motivators for the consumption of SS were unskilled personnel, [16,18,22], although in a recent work in mountain runners, these percentages were similar (20.7% advised by dietitian–nutritionist, 17.4% by trainer, 17.4% by colleagues, and 12% by friends), except in the case of a dietitian–nutritionist who was lower in open water swimmers. It is very important to know that athletes who receive information from a dietitian–nutritionist as a main source of nutritional information have better eating habits, a better understanding of nutrient periodization, and increased scientific evidence of SS consumption in its performance-enhancing effects [26].

Regarding the place of their purchase them, which can also be a determining factor for their correct use [25], the most frequent in the sample analyzed were specialized stores, followed by pharmacies, internet, and shopping centers, with these results being very similar to those reported by mountain runners [20], and which can contribute to obtaining more professional advice, and the purchase of products of higher quality, which can reduce the possibility of contamination with prohibited substances, as compared to online purchases [25,27].

The results obtained according to the AIS classification [4], with respect to the scientific evidence for each SS group, within group A, a higher average consumption of the subgroup of sports performance supplements was found in national-level swimmers (*p* = 0.027), and a trend towards statistical differences in the subgroup of medical supplements (*p* = 0.083), as well as in group C (*p* = 0.051); but not in the subgroup of sport foods (group A), (*p* = 0.154), in the total consumption of group A (*p* = 0.197), nor in the consumption of group B (*p* = 0.642). The differences found mainly in the subgroups of group A, according to the level of competition, were similar to those previously found in other studies, and support the hypothesis that this is one of the most determinant variables in SS consumption according to the level of evidence [17,18]. However, it is necessary to indicate that the consumption of SS belonging to group C of the AIS classification [4] (little or no evidence of beneficial effects) is similar to group A (maximum evidence of improved performance), and much higher than those of group B (need further research), perhaps due to high consumption of group C SS, such as BCAAs or glutamine in both groups of swimmers. Therefore, it is important to provide nutrition education to athletes that can help them make better use of supplements [25,28], as many athletes frequently consume SS without understanding its effects or risks [25].

As for the SS that were most consumed by the sample studied, we found sports drinks (62.9%), energy bars (53.0%), caffeine (39.4%), vitamin C (25.0%), and vitamin D (22.7%), with the first three coinciding with the most-consumed SS by mountain runners [20]. In addition, these results are similar to those reported in other studies with elite athletes in various sports modalities [3,22,23].

It is important to note that 4 of the 5 most-consumed SS belonged to group A, maximum evidence from the AIS classification [4], but no significant differences were found in their consumption between groups of swimmers of different competitive levels, nor were differences found in the most-consumed SS of group B—as it occurred with the supplements BCAAs and Taurine belonging to group C, which were more consumed by national-level swimmers.

The SS most consumed in our study belong to group A, and specifically in the sports food subgroup. Sports drinks and energy bars belong to this group and are some of the most-consumed SS by athletes of any age, sex, level, or sport [3]. These products provide energy and nutrients in a more appropriate way than normal food for athletes, constitute a simple way to ingest the right amount of macronutrients, and comply with dietary-nutritional recommendations for competition [2]. In addition, sports drinks are a good source of fluids, sodium, and carbohydrates, both during and after exercise, helping with the rehydration of the athlete and loading [29]; similarly, sports bars can be a good source of carbohydrates during and after exercise, providing carbohydrates, proteins, and micronutrients [2,4,19,30]. In addition, these adequate intakes of nutrients can prevent health problems during competitions, such as dehydration, which can be prevented by the intake of between 400–800 mL of liquids per hour of competition [31]; hyponatremia, which can be prevented by taking 300–600 mg of sodium per hour of competition [31]; or by emptying muscle glycogen, by taking 60–90 g/h of CH in competitions of up to 2 h, or at least 90 g/h in those lasting more than 2.5 h [32]; for all these objectives, sport foods can be a great option, as there is no possibility for athletes to bring their own food during a competition, and must consume it at the refreshment stations, since, in OWS, swimmers must rely on self-fueled fuel and fluid sources, using feeding zones only when tactically appropriate [8].

Caffeine was the third most-consumed SS by open water swimmers, but no significant differences were found according to the competitive level, similar to the result reported in other sports modalities [22,23,33]. Caffeine supplementation has been shown to increase alertness and improve performance, reduce the rate of perceived exertion (RPE), improve cognitive performance, and improve muscle energy during exercise [33]. The ergogenic effect of caffeine on sports performance is similar in elite and lower-level athletes [34], making this SS suitable for open water swimmers of all levels.

Although SS consumption is widespread and standardized, it is necessary for both health professionals and athletes to make a cost–benefit analysis prior to their use [25], based on their safety, efficiency, and legality [35]. In addition, the consumption of SS should be a supplement to the planning of the athlete and its use does not compensate for an inadequate diet or a poor choice of food [25].

As for the limitations of the present study, we can say that, although the sample of 132 swimmers was low, it is representative with respect to the number of federal licenses of swimmers in open water in Spain, representing 33.2% of the 397 federated swimmers (152 women), in the 2019/2020 and 2020/2021 seasons in which the study was conducted. In addition, although the tool to evaluate SS consumption in athletes was a validated questionnaire, the collection of this information in a self-reported and retrospective manner can induce errors in the number and type of SS consumed. No distinction was made between SS consumption by sex, given the women sample size, and because they all belonged to the same competitive level, which did not allow for obtaining differences with sufficient statistical significance. Although the sample was limited, this is the first study that analyzed supplementation specifically in open water swimmers. To overcome these limitations, as future research lines, it is proposed to extend the data collection to the total of federated open water swimmers in Spain, including a big sample of woman to carry out analysis according to sex differences. In addition, efforts can be made to collaborate with federations in other countries and thus have a more representative SS consumption pattern worldwide and check whether the SS consumption is similar in all of them.

## 5. Conclusions

We can conclude that most of the open water swimmers consume SS (around 80%), being higher in those in a higher competitive level. The majority consumption pattern includes sports foods (sports drinks, energy bars), performance supplements (caffeine), and medical supplements (vitamins C and D). All of them belong to the category of most scientific evidence. Although its consumption is minor (<20%), some swimmers consume supplements with scientific evidence not supportive of benefits among athletes or no research under-taken to guide an informed opinion (Group C by AIS). In addition, the advice, although in a low percentage, but greater to similar studies, was completed by a dietitian–nutritionist, together with the place of purchase of these SS, may have contributed to this increased consumption of SS of greater evidence. Finally, this work obtained results similar to other studies that analyzed supplementation in other sports. These results provide the first evidence of SS consumption in Spanish open water swimmers and demonstrate some specific patterns according to the high demands of volume and intensity in the swimming training programs.

## Figures and Tables

**Table 1 nutrients-14-05211-t001:** Descriptive data of the open water swimmers at the regional and national level.

Competition Level	Sex	Age (Years)	Height (cm) *	Weight (kg) *	BMI (kg·m^−2^) *	Experience (Years)
National (*n* = 67)	Men (*n* = 67)	44.13 ± 12.20	178.46 ± 5.69	80.46 ± 10.79	25.22 ± 2.8	5.93 ± 2.99
Regional (*n* = 65)	Men (*n* = 36)	43.28 ± 14.30	179.61 ± 6.00	81.07 ± 13.77	25.09 ± 3.93	5.75 ± 2.99
	Women ** (*n* = 29)	40.52 ± 12.42	165.97 ± 6.81	62.34 ± 8.72	22.62 ± 2.77	5.69 ± 3.17

Data presented as M ± SD; * Self-reported height and weight. BMI calculated from self-reported height and weight. ** All the women participated at the regional level.

**Table 2 nutrients-14-05211-t002:** Number of SS consumed by national and regional level swimmers in the specific category established by the AIS [4].

Group	Subcategory	Total Sample	National Level	Regional Level
Group A	Sport foods	1.90 ± 1.41	2.07 ± 1.47	1.72 ± 1.34
	Medical supplements	1.24 ± 1.48	1.46 ± 1.70	1.02 ± 1.19
	Sport performance *	0.70 ± 0.97	0.88 ± 1.14	0.51 ± 0.73
Group B		0.45 ± 0.76	0.48 ± 0.68	0.42 ± 0.85
Group C		1.88 ± 2.61	2.31 ± 2.95	1.43 ± 2.14

Data presented as M ± SD; * Statistical differences (*p* < 0.05) between national and regional level swimmers.

**Table 3 nutrients-14-05211-t003:** SS consumption of national and regional level swimmers in the specific category established by AIS [4].

Group	Subcategory	Supplement	Total Sample	National Level	Regional Level	*p*-Value
Group A	Sport foods	Sport drinks	62.9%	68.7%	56.9%	0.208
		Energy bars	53.0%	56.7%	49.2%	0.486
		Electrolytes	16.7%	16.4%	16.9%	1.00
		Whey protein	15.9%	11.9%	20.0%	0.240
		Gainers	10.6%	10.4%	10.8%	1.000
	Medical supplements	Vitamin D	22.7%	22.4%	23.1%	1.000
		Vitamin complex	18.9%	19.4%	18.5%	1.000
	Sport performance *	Caffeine	39.4%	46.3%	32.3%	0.112
		Creatine Monohydrate	12.9%	13.4%	12.3%	1.000
Group B		Vitamin C	25.0%	26.9%	23.1%	0.690
		Carnitine	12.1%	16.4%	7.7%	0.182
		Magnesium	22.0%	25.4%	18.5%	0.403
		Vitamin E	11.4%	16.4%	6.2%	0.098
Group C		BCAA *	19.7%	26.9%	12.3%	0.048
		Glutamine	15.9%	17.9%	13.8%	0.636
		Green tea	11.4%	13.4%	9.2%	0.585
		Royal jelly	10.6%	10.4%	10.8%	1.000
		Taurine *	10.6%	17.9%	3.1%	0.009

* Statistical differences (*p* < 0.05) between national and regional level swimmers.

## Data Availability

The data presented in this study are available in the tables of this article. The data presented in this study are available on request from the corresponding author.

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
