# Peer review of "Analysis of the Consumption of Sports Supplements in Open Water Swimmers According to the Competitive Level"

_nutrients, 2022, doi:10.3390/nu14245211_

Round 1
Reviewer 1 Report
The article is trying to analysis of the consumption of sports supplements in open water swimmers according to the competitive level. However, the innovation were not well addressed.
Issue 1. Keywords are not refined enough, especially the last three keywords.
Issue 2. The writing level of the article is not good: English tense is not allowed, the sentence is not smooth, small sentences should be stacked to show logic.
eg.”The 79.5% of the participants consumed SS, with significant differences according to their 27 level, with a higher consumption found in athletes at the national level.” The two "the" are a bit repetitive; The second "with" sentence should be the explanation of the first. Think about it.
eg.“between 40 and 70%”The article is lack of discipline in writing. Not only is "between and" incorrect, the percent sign should come after each digit.
Issue 3. The aesthetics of the table need to be improved. The line thickness of the table is not adjusted properly(Table 1),and it is not recommended that a table be spread across pages(Table 3.).
Issue 4. The conclusions of this study are similar to those of other studies analyzing other sports supplements. Although there are some innovations in specific research directions, there is no substantial progress from the conclusion.
Issue 5.The purpose of this study is to analyze the relationship between the consumption patterns of open water swimmers and the choice of 67 sports supplements and the scientific degree 68 evidence and competition level. We don't see a specific consumption pattern in the conclusion and the conclusion is incomplete.
Author Response
The article is trying to analysis of the consumption of sports supplements in open water swimmers according to the competitive level. However, the innovation were not well addressed.
Issue 1. Keywords are not refined enough, especially the last three keywords.
Response of the authors: According to the reviewer's suggestions, the last three keywords have been changed and improved.
Issue 2. The writing level of the article is not good: English tense is not allowed, the sentence is not smooth, small sentences should be stacked to show logic.
eg.”The 79.5% of the participants consumed SS, with significant differences according to their 27 level, with a higher consumption found in athletes at the national level.” The two "the" are a bit repetitive; The second "with" sentence should be the explanation of the first. Think about it.
eg.“between 40 and 70%”The article is lack of discipline in writing. Not only is "between and" incorrect, the percent sign should come after each digit.
Response of the authors: According to the reviewer's suggestions, English has been reviewed and some phrases have been improved for a better comprehension.
Issue 3. The aesthetics of the table need to be improved. The line thickness of the table is not adjusted properly (Table 1),and it is not recommended that a table be spread across pages(Table 3.)
Response of the authors: According to the reviewer's suggestions, Table 1 has been adjusted properly and Table 3 has been replaced on a new page.
Issue 4. The conclusions of this study are similar to those of other studies analyzing other sports supplements. Although there are some innovations in specific research directions, there is no substantial progress from the conclusion.
Response of the authors: The conclusion section has been modified to include the reviewer's suggestions
Issue 5.The purpose of this study is to analyze the relationship between the consumption patterns of open water swimmers and the choice of 67 sports supplements and the scientific degree 68 evidence and competition level. We don't see a specific consumption pattern in the conclusion and the conclusion is incomplete.
Response of the authors: The conclusion section has been modified to include the reviewer's suggestions

Reviewer 2 Report
The aim of this work is to analyse the pattern of consumption of SS by open water swimmers, by studying the differences according to the competitive level. It is an interesting topic and the research is generally well presented in the manuscript.
Some improvements are suggested:
- To clarify in the introduction section what is the contribution to what is already known about this manuscript
- To better describe participants' recrutiment
- To improve the questionnaire's description
- To improve the contribution of this research to the field in the discussion section
- To describe in depht the limitations and possible solutions and future research lines.
Author Response
The aim of this work is to analyse the pattern of consumption of SS by open water swimmers, by studying the differences according to the competitive level. It is an interesting topic and the research is generally well presented in the manuscript.
Some improvements are suggested:
- To clarify in the introduction section what is the contribution to what is already known about this manuscript
Response of the authors: According to the reviewer's suggestions the contribution of this work to knowledge has been included in the introduction.
- To better describe participants' recrutiment
Response of the authors: According to the reviewer's suggestions the recruitment process for participants has been further detailed.
- To improve the questionnaire's description
Response of the authors: According to the reviewer's suggestions, the description of the questionnaire has been further elaborated.
- To improve the contribution of this research to the field in the discussion section
Response of the authors: According to the reviewer's suggestions, the first part of discussion section has been modified to highligth the contribution of this research.
- To describe in depht the limitations and possible solutions and future research lines.
Response of the authors: According to the reviewer's suggestions consideration has been included for mitigating limitations as well as possible future lines of research.

Reviewer 3 Report
Dear Authors,
Thank you for the opportunity to review your manuscript. Please see below for my comments on each section.
Introduction:
Line 48: The phrasing is a little difficult to understand in this line, would you please be able to re-phrase?
Line 50: Could you please provide some more detail as to the physiological challenges associated with open water swimming events?
Line 56: Again, could some more detail be provided as to the challenges associated with the nutrients being supplied/stored on/within floating stations (e.g., pacing challenges)?
Line 57: I would suggest re-phrasing as 'in endurance sports...' or similar.
Line 58: Please provide a brief explanation as to potential mechanisms by which caffeine can elicit an improvement in performance.
Lines 62-64: I would suggest for this section some re-phrasing to state what the previous study investigated and found, and what the current paper will add to the research area.
Line 67: Within the Aim, I would request some further detail as to the types of analysis and units to be used to answer the research question, and for clearer information as to whether this is an original investigation or review article, etc.
Materials and Methods:
Lines 72-73: I would suggest a slight re-phrasing, such as 'a descriptive and cross-sectional study was used to investigate the consumption and regular use of SS of open water swimming', or similar.
Line 75: Would this refer to N = 0.05 or p = 0.05?
Line 85: Please include an indication of the error values presented here - this is - are these mean and standard deviation values?
Line 95: Could you please provide some examples and details of the different types of information collected (anthropometric, personal and social)? Could the same also be provided for the other categories (sports and SS consumption)?
Results:
Line 122: I would suggest to make it clear within this statement that the 79.5% refers to the whole sample of athletes within the study.
Line 134-135: Does this statement mean that there were no statistical differences in athletes' use of supplements in category A compared with supplements in category B?
Line 148: Does this refer to a the potential for performance benefit with the use of supplements?
Line 156: I would suggest to make the information in this table clearer within the text, so that the reader can understand the classification of the swimmers (e,g. 67 national level competitors (67 males, 0 females) and 65 regional level competitors (36 males, 29, females).
Discussion:
Line 170: I would also suggest to acknowledge the broader scope of studies that have investigated SS in other sports (i.e., other than endurance sports).
Line 173: Please specify here that you are referring to the current investigation (so that the reader understands the differentiation between the current investigations and previous studies).
Line 176: Could you please indicate (with the previous study) the types of sports/athletes this previous study on Spanish athletes focused on.
Line 182 (and above): I would suggest with this section to focus on the key findings in the first paragraph, and then avoid any further re-statement of results in the text below.
Line 188: Similar to my point above, this reads more like a statement of results, rather than an interpretation of results in the context of the existing literature; please re-phrase.
Line 204: This paragraph also reads more like a re-statement of results rather than an interpretation of results.
Line 221: Suggest to combine this paragraph with the one above.
Line 239: This paragraph provides valid points, but I would suggest stating the most relevant result from the current study at the beginning of the paragraph, so that relevant points from previous research can then be incorporated into the paragraph.
Author Response
Dear Authors,
Thank you for the opportunity to review your manuscript. Please see below for my comments on each section.
Introduction:
Line 48: The phrasing is a little difficult to understand in this line, would you please be able to re-phrase?
Response of the authors: According to the reviewer's suggestions, the phrase has been improved for a better comprehension.
Line 50: Could you please provide some more detail as to the physiological challenges associated with open water swimming events?
Response of the authors: According to the reviewer's suggestions the explanation of the physiological challenges of this sport discipline has been included.
Line 56: Again, could some more detail be provided as to the challenges associated with the nutrients being supplied/stored on/within floating stations (e.g., pacing challenges)?
Response of the authors: According to the reviewer's suggestions, more information about the process of obtaining nutrients in open water swimmers has been added.
Line 57: I would suggest re-phrasing as 'in endurance sports...' or similar.
Response of the authors: According to the reviewer's suggestions, the phrase has been changed to “In endurance sports…”.
Line 58: Please provide a brief explanation as to potential mechanisms by which caffeine can elicit an improvement in performance.
Response of the authors: The authors appreciate the reviewer's comments, but this information is in the discussion section and we believe that adding it to the introduction would extend the manuscript. In addition, this would lead to the explanation of the effect of the rest of the sports supplements mentioned in this section.
Lines 62-64: I would suggest for this section some re-phrasing to state what the previous study investigated and found, and what the current paper will add to the research area.
Response of the authors: According to the reviewer's suggestions, this section has been rewritten.
Line 67: Within the Aim, I would request some further detail as to the types of analysis and units to be used to answer the research question, and for clearer information as to whether this is an original investigation or review article, etc.
Response of the authors: According to the reviewer's suggestions, more information about the study has been added to the aim and material and methods section has been improved to clarify the methodology of the study.
Materials and Methods:
Lines 72-73: I would suggest a slight re-phrasing, such as 'a descriptive and cross-sectional study was used to investigate the consumption and regular use of SS of open water swimming', or similar.
Response of the authors: According to the reviewer's suggestions, the text has been revised and re-phrased.
Line 75: Would this refer to N = 0.05 or p = 0.05?
Response of the authors: The mistake has been corrected in the manuscript.
Line 85: Please include an indication of the error values presented here - this is - are these mean and standard deviation values?
Response of the authors: According to the reviewer's suggestions the indication has been included.
Line 95: Could you please provide some examples and details of the different types of information collected (anthropometric, personal and social)? Could the same also be provided for the other categories (sports and SS consumption)?
Response of the authors: According to the reviewer's suggestions, examples of all paragraphs developed in the questionnaire have been included.
Results:
Line 122: I would suggest to make it clear within this statement that the 79.5% refers to the whole sample of athletes within the study.
Response of the authors: According to the reviewer's suggestions, the phrase has been modified.
Line 134-135: Does this statement mean that there were no statistical differences in athletes' use of supplements in category A compared with supplements in category B?
Response of the authors: This statement means that no differences were found according to competition level neither in category A nor category B. To clarify the text, this statement has been rewritten.
Line 148: Does this refer to a the potential for performance benefit with the use of supplements?
Response of the authors: Yes, the athletes hope to obtain an improvement in sports performance through the consumption of the supplements. These answers are predetermined in the questionnaire based on scientific reference documents in sports nutrition (e.g.: doi: 10.1136/bjsports-2018-099027). Athletes select the answer they consider most appropriate.
Line 156: I would suggest to make the information in this table clearer within the text, so that the reader can understand the classification of the swimmers (e,g. 67 national level competitors (67 males, 0 females) and 65 regional level competitors (36 males, 29, females).
Response of the authors: To make the table 1 clearer has been moved to material and methods section near the text that describes it (subsection 2.2.).
Discussion:
Line 170: I would also suggest to acknowledge the broader scope of studies that have investigated SS in other sports (i.e., other than endurance sports).
Response of the authors: According to the reviewer's suggestions, this paragraph has been modified.
Line 173: Please specify here that you are referring to the current investigation (so that the reader understands the differentiation between the current investigations and previous studies).
Response of the authors: According to the reviewer's suggestions, these lines have been modified.
Line 176: Could you please indicate (with the previous study) the types of sports/athletes this previous study on Spanish athletes focused on.
Response of the authors: According to the reviewer's suggestions, the type of sport/athletes included in the study has been specified.
Line 182 (and above): I would suggest with this section to focus on the key findings in the first paragraph, and then avoid any further re-statement of results in the text below.
Response of the authors: The reviewer's suggestions have been included in the manuscript.
Line 188: Similar to my point above, this reads more like a statement of results, rather than an interpretation of results in the context of the existing literature; please re-phrase.
Response of the authors: The reviewer's suggestions have been included in the manuscript.
Line 204: This paragraph also reads more like a re-statement of results rather than an interpretation of results.
Response of the authors: The reviewer's suggestions have been included in the manuscript.
Line 221: Suggest to combine this paragraph with the one above.
Response of the authors: According to the reviewer's suggestions, the paragraph has been combined with the one above.
Line 239: This paragraph provides valid points, but I would suggest stating the most relevant result from the current study at the beginning of the paragraph, so that relevant points from previous research can then be incorporated into the paragraph.
Response of the authors: The reviewer's suggestions have been included in the manuscript.

Round 2
Reviewer 1 Report
The article is trying to analysis of the consumption of sports supplements in open water swimmers. It has been revised according to the suggestions. However, some issues were not well addressed.
Issue 1: As shown in Table 1, why the women swimmer were not included in the National group?
Issue 2: As it was described in the part 2.3 Instruments, the questionnaire was the important instrument in this research, the author should provide the questionnaire as the Supplementary data.

Reviewer 2 Report
Dear Authors,
You have elaborated an answer to my every comment and you have significantly improved the quality of your manuscript.
Author Response
Thanks you very much for your comments.